# The Nutrients and Volatile Compounds in *Stropharia rugoso-annulata* by Three Drying Treatments

**DOI:** 10.3390/foods12102077

**Published:** 2023-05-22

**Authors:** Yu Jiang, Qilong Zhao, Haolan Deng, Yongjun Li, Di Gong, Xiaodan Huang, Danfeng Long, Ying Zhang

**Affiliations:** 1School of Public Health, Lanzhou University, Lanzhou 730000, China; jiangy20@lzu.edu.cn (Y.J.); zhaoql21@lzu.edu.cn (Q.Z.); denghl18@lzu.edu.cn (H.D.); gongd@lzu.edu.cn (D.G.); huangxiaodan@lzu.edu.cn (X.H.); longdf@lzu.edu.cn (D.L.); 2Gansu Provincial Center for Disease Control and Prevention, Lanzhou 730000, China; 13893203870@126.com

**Keywords:** *Stropharia rugoso-annulata*, drying methods, nutrients, volatile compounds, sensory evaluation

## Abstract

This study aimed to examine the differences in the nutrients and volatile compounds of *Stropharia rugoso-annulata* after undergoing three different drying treatments. The fresh mushrooms were dried using hot air drying (HAD), vacuum freeze drying (VFD), and natural air drying (NAD), respectively. After that, the nutrients, volatile components, and sensory evaluation of the treated mushrooms were comparably analyzed. Nutrients analysis included proximate compositions, free amino acids, fatty acids, mineral elements, bioactive compositions, and antioxidant activity. Volatile components were identified by headspace-solid phase microextraction-gas chromatography-mass spectrometry (HS-SPME-GC-MS) and analyzed with principal component analysis (PCA). Finally, sensory evaluation was conducted by ten volunteers for five sensory properties. The results showed that the HAD group had the highest vitamin D_2_ content (4.00 μg/g) and antioxidant activity. Compared with other treatments, the VFD group had higher overall nutrient contents, as well as being more preferred by consumers. Additionally, there were 79 volatile compounds identified by HS-SPME-GC-MS, while the NAD group showed the highest contents of volatile compounds (1931.75 μg/g) and volatile flavor compounds (1307.21 μg/g). PCA analysis suggested the volatile flavor compositions were different among the three groups. In summary, it is recommended that one uses VFD for obtaining higher overall nutritional values, while NAD treatment increased the production of volatile flavor components of the mushroom.

## 1. Introduction

*Stropharia rugoso-annulata* (*S. rugoso-annulata*), also known as the wine red mushroom, is an edible fungus highly recommended to be cultivated in developing countries by the Food and Agriculture Organization (FAO) [1]. This mushroom boasts a wealth of valuable nutrients such as protein, free amino acids, and vitamin and mineral elements [2]. Moreover, it is rich in bioactive components, including polysaccharides, phenolics, and flavonoids [3], which are reputed to confer a plethora of health benefits, such as antioxidation, anti-tumor activity, the regulation of serum lipid and glucose levels, and the promotion of gut health [4,5]. Moreover, *S. rugoso-annulata* is sought after by consumers for its pleasant aroma, which consists of various volatile compounds contributing to the overall sensory properties [6].

After it is harvested, fresh *S. rugoso-annulata* is quickly perishable, due to its high water content and active postharvest respiration, which limits the shelf life and quality preservation of the mushroom. Therefore, postharvest processing techniques, such as drying, edible film, packaging, and irradiation, are required in order to extend the shelf life and maintain the postharvest quality of *S. rugoso-annulata* [7,8]. Drying is one of the most efficient and convenient techniques in the food industry, which decreases the moisture of the mushroom to a level below 10% fresh weight so as to enable long-term storage and make the mushroom available out of the harvest seasons [9]. Hot air drying (HAD), vacuum freeze drying (VFD), and natural air drying (NAD) are the three most common drying methods for edible mushrooms [10]. HAD is a fast and cost-effective drying method, but it has the disadvantages of leading to nutrient loss and the quality reduction of mushrooms [11]. NAD is an environmentally friendly method, which can preserve the natural flavor and quality of mushrooms. However, it is relatively slow and could be affected by weather conditions [12]. Moreover, VFD has the advantages of maintaining the natural color, flavor, and nutrients of mushrooms. Nevertheless, it is expensive and requires specialized equipment [13]. In general, the choice of drying method depends on the specific product and the intended result.

Previous research suggested that drying methods had different impacts on the nutrients and volatile components of mushrooms [14]. Yang et al. [15] discovered that freeze drying (FD) was the most suitable method for preserving protein and free amino acids, and reducing sugar in fresh *Pleurotus eryngii*, compared with NAD and HAD. Moreover, FD treatment enhances the production of umami flavor in dried *Lentinula edodes*, while it causes a higher release of the typical shiitake mushroom aroma found in HAD-treated mushroom [16]. In addition, Liu et al. [17] reported that VFD treatment significantly increased the total volatile compounds of fresh *Lentinula edodes*, compared with HAD treatment. However, with regard to *S. rugoso-annulata*, previous research only studied the effects of drying methods on the non-volatile compounds, which showed that VFD-treated samples had the highest contents of free amino acids, organic acids, and soluble sugars [18]. Few studies researched the effects of different drying methods on the nutrients and volatile compositions in *S. rugoso-annulata*. With the increase of consumers’ health awareness, the development of processed products with high nutrients and sensory quality has attracted wide attention. *S. rugoso-annulata* is considered as a kind of edible fungus with abundant nutrients and volatile compounds, so it is of great value to study the effects of drying treatments on its nutrients and volatile compounds.

Therefore, the objective of this study is to evaluate the effects of HAD, VFD, and NAD on the nutrients and volatile components of *S. rugoso-annulata*. The results may contribute to developing the optimal postharvest technique to maintain the quality of *S. rugoso-annulata*.

## 2. Materials and Methods

### 2.1. Mushroom Samples

Fresh *S. rugoso-annulata* at the same growth stage (with similar length and unopened cap) was harvested from a local farm in Linxia, Gansu Province (35°28′5″ N, 103°24′50″ E). All fresh mushrooms were transferred to the laboratory on the same day, then cleaned with distilled water. After drying the surface with flowing air, 30 g of the fresh mushroom was dried at 105 ± 1 °C to measure the moisture of fresh mushroom, while the remaining mushrooms (15 kg) were cut into 5 mm slices.

### 2.2. Drying Methods

#### 2.2.1. Hot Air Drying (HAD)

For hot air drying, 5 kg of the fresh mushroom slices were dried at 65 °C for 10 h with electric constant-temperature drying oven (DHG-9055A, Blue Pard, Shanghai, China) until final moisture content was below 10% fresh weight.

#### 2.2.2. Vacuum Freeze Drying (VFD)

For vacuum freeze drying, 5 kg of the fresh mushroom slices were frozen at −20 °C for 24 h, and then dried by a vacuum freeze dryer (FD-1A-50, BIOCOOL, Beijing, China) with the vacuum degree of 20–40 Pa. The temperature of cold trap and drying chamber were −50 °C and −35 °C. The drying process lasted for 48 h until remaining moisture content was under 10% fresh weight.

#### 2.2.3. Natural Air Drying (NAD)

For natural air drying, 5 kg of the fresh mushroom slices were dried with natural air (25–30 °C) for 72 h until moisture content was under 10% fresh weight. The average relative humidity of environment was 50%.

### 2.3. Proximate Compositions Analysis

The samples were dried at 105 ± 1 °C to measure the moisture; crude fat and ash were analyzed by the AOAC methods; and crude protein was determined with Kjeldahl method by N × 4.38 [19,20]. Total carbohydrate was calculated as follow. All the results were expressed as percent of dry weight.
Total carbohydrate (%) = 100 − (ash + fat + moisture + protein)(1)

### 2.4. Free Amino Acids Analysis

Free amino acid compositions were detected using amino acid analyzer according to the method of Fu et al. [21]: 0.1 g of mushroom was subjected to hydrolysis using 6 mol/L hydrochloric acid (10 mL) at 110 °C for 22 h. The mixture was centrifuged, and then the supernatant was collected and dried at 50 °C before being redissolved in distilled water. The process was repeated twice to remove any remaining hydrochloric acid. Subsequently, the residue was dissolved in sodium citrate buffer solution at a pH of 2.2. After filtering with 0.22 μm Millipore membrane, the sample was analyzed by an amino acid analyzer (S-433D, SYKAM, Eresing, Germany). The results were expressed as g/100 g of dried mushroom.

### 2.5. Fatty Acids Analysis

Fatty acids were analyzed, complying with the method of Amusquivar et al. [22]: 3 g of mushroom was weighed into a test tube, then mixed with 10 mL methylbenzene and 15 mL 5% acetyl chloride methanol solution. After that, the tube was sealed with polyethylene cap and kept in a water bath at 70 °C for 2 h. When cooling to room temperature, 5 mL potassium carbonate (6%) and 2 mL methylbenzene were added into the mixture, then mixed by vortex. The sample was centrifuged at 6000 rpm for 5 min, then the upper organic phase was transferred into a new tube. Subsequently, anhydrous sodium sulfate and active carbon were added to remove the residual water, and the supernatant was collected and filtered through 0.22 μm nylon membrane. The contents of fatty acids were measured by a gas chromatography tool equipped with flame ionization detection (GC-FID, 7890A, Agilent Technologies, Santa Clara, CA, USA). The spectrum column was SP-2056 column (100 m × 0.25 mm × 0.2 μm, SUPELCO, St. Louis, MO, USA). The temperature procedure was initiated at 70 °C, held for 4 min, then increased at a rate of 8 °C/min up to 110 °C for 2 min, followed by a ramp to 170 °C at a rate of 5 °C/min, which was maintained for 10 min, finally reaching 255 °C at a rate of 4 °C/min for 45 min. The carrier gas was hydrogen with a flow rate of 1 mL/min. The inject volume was 2 μL with a split ratio of 1:50, and the injector temperature was 255 °C. Fatty acids were identified by comparing the relative retention times of samples with standards. The content of fatty acid was expressed as percentage.

### 2.6. Mineral Elements Analysis

Mineral elements were analyzed following previous work of Falandysz et al. [23]. In brief, 200 mg of mushroom powder was added into a 10 mL glass tube with 2 mL of ultrapure nitric acid (65%) and soaked overnight. After that, the mixture was digested for 33 min in microwave digestion system (Ultra Wave, Milestone, Milan, Italy) with the temperature increasing from 25 °C to 130 °C at 21 °C/min which was kept for 3 min, followed by an increase to 200 °C at 14 °C/min which was kept 20 min. After digestion, the liquid was filtered using 0.45 μm Millipore filter. All elements in both samples and standards were detected using inductively coupled plasma mass spectrometry (ICP-MS) (Plasma Quant MS, Analytik, Jena, Germany) in triplicate. The power of plasma was 1.35 kW, and the speeds of peristaltic pump, cool gas, aux gas, and nebulizer gas were 15 r/min, 10.5 L/min, 1.35 L/min, and 0.94 L/min, respectively. All gases were argon. The contents of mineral elements were expressed as mg/kg of dry weight.

### 2.7. Health Risk and Daily Intake Contribution Evaluation

To analyze the potential health risk of *S. rugoso-annulata*, the estimated daily intake (EDI) and target hazard quotient (THQ) of potentially toxic trace elements (Cd, Hg, Pb, and As) were calculated. EDI and THQ were calculated with the following equations:(2)$$\mathrm{EDI}\;(\mathrm{mg}/\mathrm{kg})=\frac{C\times \mathrm{IR}\times \mathrm{EF}\times \mathrm{ED}}{\mathrm{ET}\times \mathrm{BW}}$$
where EF stands for intake frequency (365 days/year), ED for intake years (30 years), ET for average intake time (365 days/year × 30 years), and BW for mean adult body weight (60 kg) [24]. C is the mean element concentration in *S. rugoso-annulata* (mg/kg), and IR is the mushroom intake rate (6.6 × 10^−3^ kg/person/day) [25].
(3)$$\mathrm{THQ}=\frac{\mathrm{EDI}}{R_{f}D}$$
where R_f_D (mg/kg/day; kg refers to kg of body weight) denotes the oral reference dosage of element, and the R_f_D values for Cd, Hg, Pb, and As are 0.001, 0.0003, 0.004, and 0.0003 mg/kg/day, respectively [25].
(4)$${\mathrm{TI}=\mathrm{THQ}}_{\mathrm{Cd}}{+\mathrm{THQ}}_{\mathrm{Hg}}{+\mathrm{THQ}}_{\mathrm{Pb}}{+\mathrm{THQ}}_{\mathrm{As}}$$

Toxic index (TI) is the sum of four THQ values. If the TI is lower than 1, this mushroom is considered to be safe for long-term consumption. On the contrary, there may be potential adverse health effects.

To analyze the daily intake contribution of *S. rugoso-annulata*, the percentage contribution of essential trace elements (Fe, Cu, Zn, Se, Cr, and Co) to recommended daily intake (RDA) was calculated with the following equations. The RDA values of Fe, Cu, Zn, Se, Cr, and Co were 8, 0.9, 11, 0.055, 0.035, and 0.0001 mg/kg [26,27].
(5)$$\mathrm{Percentage}\;\mathrm{contribution}\;(\%)=\frac{C\times \mathrm{IR}}{\mathrm{RDA}}$$

### 2.8. Bioactive Compositions and Antioxidant Activity Analysis

#### 2.8.1. Polysaccharides

Crude polysaccharides were isolated and detected based on the method of Akram et al. [28] with slight changes. *S. rugoso-annulata* was extracted with distilled water (1:25 g/mL) at 60 °C for 1 h. Subsequently, the mixture was centrifuged at 8000 rpm for 20 min to obtain the supernatant. The sediment was extracted twice more, and the supernatants were combined and precipitated overnight at 4 °C with fourfold the amount of ethanol. After that, the solution was centrifuged at 8000 rpm for 20 min to separate precipitate, which was subsequently dissolved in distilled water. The content of crude polysaccharides was measured using phenol–sulfuric acid method and expressed as g/100 g of dried mushroom. 

#### 2.8.2. Total Phenolics and Total Flavonoids

Following the method of Li et al. [29] with some minor adjustments, 0.5 g of the sample was dissolved in 10 mL of ethanol and extracted twice with ultrasonic at 50 °C, 200 W (KQ-600DE, Kunshan Ultrasonic Wave Technology Corporation, Kunshan, China), in the dark for 10 min, and then centrifuged at 8000 rpm for 10 min. The supernatants were combined and diluted to a final volume of 25 mL with ethanol. Total phenolics were analyzed using Folin–Ciocalteu method with a standard of gallic acid. The absorbances of both sample and standard were measured by an ultraviolet-visible spectrophotometer (UV-759, YOKE, Shanghai, China) at 725 nm, and the total phenolic content (TPC) was expressed as mg gallic acid equivalent per gram (mg GAE/g). Total flavonoids were analyzed using AlCl_3_-NaNO_2_-NaOH method with a standard of rutin. The absorbance was measured by an ultraviolet-visible spectrophotometer (UV-759, YOKE, Shanghai, China) at 510 nm, and the total flavonoid content (TFC) was presented as mg rutin equivalent per gram (mg RE/g).

#### 2.8.3. Ergosterol, Vitamin D_2_, and α-Tocopherol

Ergosterol, vitamin D_2_, and α-tocopherol were analyzed using HPLC according to the method of Liu et al. [30] with some modifications. First, 1 g of sample was mixed with 15 mL of ascorbic acid (100 g/L), then extracted with 40 mL of ethanol, 15 mL of methanol and 10 mL of potassium hydroxide solution (50%) at 80 °C for 30 min. After cooling, petroleum ether was added into the mixture and shaken for 5 min, then left standing, obtaining the organic phase. The above process was repeated twice. Subsequently, the organic phases were combined and washed to neutral with distilled water. The water phase was discarded and the residual water was removed by anhydrous sodium sulfate. Then, the organic phase was evaporated at 50 °C and dissolved using 2 mL of mobile phase. After filtering through 0.22 μm Millipore membrane, the extracted solution was tested using HPLC (1200 series, Agilent, Santa Clara, CA, USA) equipped with DAD detector. The compositions of mobile phase were acetonitrile, dichloromethane, and methanol (70:20:10, *v*/*v*/*v*) with a flow rate of 1.0 mL/min. The column was C18 column (150 × 4.60 mm, 4 μm film thickness, Phenomenex, Torrance, CA, USA) and the column temperature was 25 °C. The detecting wavelength of ergosterol, vitamin D_2_, and α-tocopherol were 282 nm, 265 nm, and 285 nm, respectively. The content of ergosterol was expressed as mg/g of dried mushroom, and the contents of vitamin D_2_ and α−tocopherol were expressed as μg/g of dried mushroom.

#### 2.8.4. β-Carotene and Lycopene

The determination was carried out following the method of Robaszkiewicz et al. [31]. First, 0.5 g of mushroom was extracted with 5 mL of methanol, followed by centrifuging at 8000 rpm for 10 min to obtain the supernatant. The absorbance was detected by ultraviolet-visible spectrophotometer (UV-759, YOKE, Shanghai, China) at 663 nm, 505 nm, and 453 nm, respectively. The following equations were used to determine the contents of β-carotene and lycopene:(6)$$\beta -\mathrm{Carotene}\;(\mathrm{mg}/100\;\mathrm{mL})=0.216A_{663}-0.304A_{505}+0{.452A}_{453}$$
(7)$$\beta -\mathrm{Carotene}\;(\mu g/g)=\frac{\beta -\mathrm{Carotene}\;(\mathrm{mg}/100\;\mathrm{mL})\times 5\times 10^{3}}{0.5}$$
(8)$$\mathrm{Lycopene}\;(\mathrm{mg}/100\;\mathrm{mL})=-0.0458A_{663}+0.3724A_{505}-0.0806A_{453}$$
(9)$$\mathrm{Lycopene}\;(\mu g/g)=\frac{\mathrm{Lycopene}\;(\mathrm{mg}/100\;\mathrm{mL})\times 5\times 10^{3}}{0.5}$$

#### 2.8.5. Antioxidant Activity

According to the method of Shomali et al. [32], 0.5 g of *S. rugoso-annulata* was extracted using 5 mL of methanol. After centrifuging at 8000 rpm for 10 min, the supernatant was collected. Subsequently, 0.1 mL of extracted solution was mixed with 3.9 mL of DPPH ethanol solution (0.2 mmol/L), and 0.1 mL of distilled water was used as control. After incubating for 30 min at room temperature in the dark, the absorbance was measured by ultraviolet-visible spectrophotometer (UV-759, YOKE, Shanghai, China) at 517 nm. The following equation was used to calculate DPPH radical scavenging rate:(10)$$\mathrm{DPPH}\;\mathrm{radical}\;\mathrm{scavenging}\;\mathrm{rate}\;(\%)=\frac{A_{C}-A_{S}}{A_{C}}\times 100$$

A_C_ and A_S_ represented the absorbance of control and sample groups.

### 2.9. Volatile Components Analysis

The detection of volatile components was conducted by headspace-solid phase microextraction-gas chromatography-mass spectrometry (HS-SPME-GC-MS) according to the method of Chen et al. [33]. *S. rugoso-annulata* was sampled by HS-SPME (AOC-6000 plus, SHIMADAZU, Kyoto, Japan). The fiber type was 80 μm DVB/PDMS/Carbon WR, which was activated according to the manufacturer’s introduction. Specifically, 0.4 g of mushroom powder was added into a headspace vial with PTFE silicon septum, then mixed with 4 mL of saturated sodium chloride solution and 15 μL of internal standard (2-octanol methanol solution, 0.1 mg/mL). The vial was kept on the heating plate for 20 min at 60 °C. 

The volatile components were qualified and quantified by GC-MS instrument (TQ8050 NX, SHIMADAZU, Kyoto, Japan) equipped with a polar capillary column (HP-VOC, 60 m × 0.2 mm × 1.12 μm, Agilent, Santa Clara, CA, USA). The injector temperature was 250 °C. The column temperature initiated at 50 °C, then raised to 90 °C by 3 °C/min, followed by 2 °C/min to 150 °C, and finally at a rate of 8 °C/min up to 250 °C where it was kept for 10 min. Helium at a rate of 1.0 mL/min was used as the carrier gas. Mass spectra ranged from 30 to 550 *m*/*z*; the ionization energy and ion source temperature were 70 eV and 250 °C, respectively. The volatile compounds in the sample were identified by comparing the mass spectra with reference standards in the NIST 14 library. The contents of volatile components were calculated according to the peak area ratio with intestinal standard, and expressed as μg/g of dried mushroom. The odor characteristics were obtained from the online databases (The LRI & Odour Database, Flavornet, and human odor space) and literature.

### 2.10. Sensory Qualities Evaluation

Sensory evaluation was carried out as described by Yang et al. [34]. Specifically, 10 volunteers (5 females and 5 males, 20–35 years old) without any dysosmia composed the sensory evaluation team. All of them had been trained by the panel in special exercises. A ten-point hedonic scale was presupposed as follows: <2, strongly dislike; 2–4, dislike; 4–6, neither like nor dislike; 6–8, like; and >8, strongly like. The sensory properties included appearance, aroma, texture, intention to buy, and overall acceptability. The sensory score for each group was expressed as a mean value.

### 2.11. Statistical Analysis

All the tests were repeated in triplicate; the results were expressed as mean ± standard deviation (SD). The statistical analysis was performed by one-way analysis of variance (one-way ANOVA) with LSD multiple range test using IBM SPSS Statistics version 24.0 (IBM, Armonk, New York, America). *p* < 0.05 was regarded as significant difference. Principal component analysis (PCA) was performed by Origin 2021 (OriginLab Corporation, Northampton, MA, USA) to demonstrate the profiles of volatile components. 

## 3. Results

### 3.1. Proximate Compositions

The moisture content of fresh *S. rugoso-annulata* was 94.04 ± 0.65%, which is consistent with the high moisture content typically found in other mushrooms and directly contributes to their short shelf life [35]. Carbohydrate was found to be the primary constituent in the dried mushroom, followed by protein (Table 1). The content of protein in dried mushrooms ranged from 17.99% to 25.67%. However, the fat content was minimal, accounting for less than ten percent of the protein content.

Edible mushrooms are renowned for their high protein content and low levels of ash and fat [36]. The lowest content of crude fat content was found in the HAD group at 0.80%, which may relate to the increase in lipase activity and oxidation of fat during the HAD process [37]. The highest protein content of *S. rugoso-annulata* showed in the VFD group, which might be due to the inhibition of proteolytic enzyme activity in the mushroom under low temperature [38].

### 3.2. Free Amino Acids

The sample was analyzed using an amino acid analyzer, and the results were presented in Table 2. Seven essential amino acids and ten non-essential amino acids were detected in the mushroom. Total and essential amino acid contents ranged from 20.3 to 29.65 g/100 g and from 6.57 to 9.72 g/100 g, respectively. Notably, the VFD group exhibited the highest total and essential amino acid contents (*p* < 0.05), consistent with its protein level. The results suggested that VFD was the optimal method for preserving amino ac ids possibly due to the inhibition of the Maillard reaction between amino acids and sugars at low temperature [39].

It is worth noting that free amino acids not only contribute to nutrition but also play a role in determining the taste properties of mushrooms. Aspartic and glutamic acids, important MSG-like ingredients, are the most essential amino acids that confer the umami characteristic of mushrooms [18]. It is noteworthy that the lowest concentration of MSG-like ingredients was 5.04 g/100 g, indicating that the level of MSG-like ingredients in *S. rugoso-annulata* was high (>2 g/100 g) [36]. Therefore, *S. rugoso-annulata* has a strong umami flavor and may gain the preference of consumers. There were significant differences in the MSG-like and sweet-taste amino acids among the three groups, with the following order of highest to lowest content: VFD > NAD > HAD. Therefore, compared with the HAD and NAD groups, VFD-treated *S. rugoso-annulata* contained higher values of essential, umami, and sweet-taste amino acids.

### 3.3. Fatty Acids

Fifteen types of fatty acids were identified in the mushroom as shown in Table 3. The most abundant fatty acids were oleic acid (C18: 1), eicosadienoic acid (C20: 2), and linoleic acid (C18: 2). The combined percentages of these three fatty acids ranged from 73.66% in the HAD group to 85.49% in the VFD group. Linoleic acid, an essential fatty acid, has been found to have numerous health benefits, such as reducing serum cholesterol and lipid levels and preventing atherosclerosis [40]. The most predominant fatty acid in *S. rugoso-annulata* was oleic acid (47.46–63.12%), which could improve postprandial insulin resistance, lower blood pressure, and reduce inflammation [41]. Similar findings were observed in *Cyathus cornucopioides* (54.02%) [35] and *Clavariadelphus pistillaris* (46.36%) [42].

Based on the different types of carbon chain connections, the fatty acids can be classified into three categories: saturated fatty acids (SFA), monounsaturated fatty acids (MUFA), and polyunsaturated fatty acids (PUFA) [43]. Among them, MUFA was found to be the most prevalent in dried *S. rugoso-annulata*, ranging from 51.79% to 66.13%, followed by PUFA (24.78–36.31%) and SFA (9.39–19.30%). A high proportion of MUFA might be associated with a reduced risk of obesity and cardiovascular diseases [44]. Our results suggested that the contents of SFA, MUFA, and PUFA were different in the three groups, with the highest percentage of SFA, MUFA, and PUFA in HAD, VFD, and NAD, respectively. Additionally, unsaturated fatty acids (UFA) predominated over SFA, as evidenced by the UFA/SFA score being greater than 1, which was consistent with wild edible mushrooms [45]. VFD had a higher UFA/SFA score than HAD and NAD, likely due to the lower temperature used during processing, resulting in a lower loss of UFA.

### 3.4. Mineral Elements Contents, Daily Intake Contribution, and Health Risk Evaluation

A total of 14 mineral elements were determined in *S. rugoso-annulata* using ICP-MS (Table 4). K, Na, Ca, and Mg are macro elements, and the mushroom contained a high level of potassium and a low level of sodium, resulting in the K/Na ratio ranging from 96.40 in the HAD group to 106.55 in the VFD group. It has been reported that the increasing dietary intake of potassium can significantly lower blood pressure in a dose-responsive manner [46], and a negative correlation has been found between the K/Na ratio and the risk of cardiovascular diseases [47]. Calcium content in *S. rugoso-annulata* varied from 359.84 to 495.92 mg/kg, which exceeded the contents in wild edible mushrooms (38–270 mg/kg) [48]. Notably, manganese is essential for normal muscular movement [49], and exhibited levels ranging from 752.35 mg/kg in the HAD group to 1049.52 mg/kg in the NAD group, surpassing the content found in wild mushrooms (84–550 mg/kg) [48].

Zn, Cu, Fe, Se, Cr, and Co are essential trace elements for human health. The absence of these elements can lead to detrimental health outcomes, including skeletal, immune, nervous, and blood system disorders [49]. Table 4 demonstrated that the VFD group had the highest concentrations of Zn (44.75 ± 2.39 mg/kg), Cu (18.72 ± 0.48 mg/kg), Se (0.47 ± 0.03 mg/kg), and Co (0.18 ± 0.00 mg/kg), whereas the NAD group had the highest Fe levels (282.07 ± 57.96 mg/kg), and the greatest amount of Cr (1.53 ± 0.10 mg/kg) showed in the HAD group (*p* < 0.05). Moreover, Table 5 showed that daily consumed *S. rugoso-annulata* can supply 10.80–23.27%, 3.28–13.73%, 0.97–2.69%, 2.18–5.67%, 12.05–28.89%, and 792.93–1164.61% to the RDA of Fe, Cu, Zn, Se, Cr, and Co, respectively. Therefore, incorporating *S. rugoso-annulata* into one’s diet can serve as a source of these crucial trace elements.

The presence of Cd, Pb, Hg, and As, as toxic trace elements, can lead to acute or chronic negative health outcomes, including neurotoxicity, liver and kidney disorders, bone damage, and cardiovascular effects [50]. The relatively low THQ values of Cd, Pb, and Hg in *S. rugoso-annulata* indicated that it may be safe for long-term consumption (Table 6). Nevertheless, the high level of arsenic in the VFD group resulted in THQ and TI scores exceeding 1, indicating possible adverse health effects. Since all the mushrooms were harvested from the same farm at the same time, moreover, all the operations except for drying treatments were the same. Therefore, the high THQ values in the VFD group might be due to sample contamination by the vacuum freeze dryer, highlighting the need to improve the VFD procedure.

### 3.5. Bioactive Compositions and Antioxidant Activity

The bioactive compositions were displayed in Figure 1, which exhibited the variations in each bioactive component in the three groups. Polysaccharides, consisting of more than ten monosaccharide units connected with glycosidic bonds, are the primary functional ingredient in *S. rugoso-annulata* [51]. Polysaccharides possess diverse health-promoting benefits, including anti-inflammatory, antioxidant, anti-cancer, and blood glucose and lipid level modulation properties [52,53]. It has been reported that finger citron fruits dried by freezing displayed the highest polysaccharides content (3.84%) compared with HAD treatment (2.74%) [54], which is consistent with our results; the polysaccharide content was higher in the VFD group with 4.04 g/100 g, followed by the HAD group with 3.90 g/100 g (Figure 1a), indicating that the two treatments could preserve the polysaccharides in *S. rugoso-annulata*.

Vitamin D has two primary forms, vitamin D_2_ and vitamin D_3_. Vitamin D_3_ is generated from 7–dehydrocholesterol due to skin exposure to ultraviolet light [55], while vitamin D_2_ and its precursor ergosterol are only found in fungi, with edible mushrooms being the most important sources [56]. Although natural mushrooms contained low levels of vitamin D_2_, UV treatment increased the concentration of vitamin D_2_ by triggering its generation from the photolysis of ergosterol [57]. Therefore, mushrooms could serve as a valuable dietary source of vitamin D for individuals lacking sunlight exposure. In this study, the VFD group had the highest level of ergosterol (1.87 ± 0.34 mg/g), while the HAD group exhibited the highest level of vitamin D_2_ (4.00 ± 0.52 μg/g). However, the NAD group had relatively low levels of both ergosterol and vitamin D_2_, which may be due to the combined effects of temperature, UV exposure, and drying time on the generation and degradation of vitamin D_2_ [58].

Total phenolics, total flavonoids, lycopene, β-carotene, and α-tocopherol contribute to the antioxidant activity of *S. rugoso-annulata* [4]. Our results indicated that the HAD group had the highest levels of total phenolics, total flavonoids, lycopene, β-carotene, and DPPH scavenging rate, indicating that the HAD-treated mushroom had higher antioxidant activity. This may be because the high temperature promoted the release of phenolic acids and flavonoid substances in the form of binding. In addition, the higher temperature may cause the inactivation of polyphenol oxidase, thereby reducing the loss of phenolic substances [59]. However, α-tocopherol was not detected in the HAD group (Figure 1d), which may be due to the degradation under the high temperature condition [60]. 

Overall, *S. rugoso-annulata* supplied a valuable source of dietary vitamin D, and HAD and VFD treatments could effectively preserve polysaccharides in the mushroom. Moreover, the HAD-treated mushroom displayed higher antioxidant activity compared with other treatments.

### 3.6. Volatile Components

A total of 79 volatile compounds were detected by HS-SPME-GC-MS analysis, comprising 17 alcohols, 18 aldehydes, 11 ketones, 6 acids, 12 esters, nine hydrocarbons, two phenols, and four other volatile compounds (Table 7). The aroma characteristics were obtained from the online database and literature. Drying treatments significantly affected the volatile compounds, with the highest values of the total volatile (1931.75 ± 90.18 μg/g) and volatile flavor compounds (1307.21 ± 47.60 μg/g) in the NAD group, followed by the VFD and HAD groups. This may be because the mushrooms are exposed to a suitable environment such as kindly temperature and humidity during the NAD treatment process, which promoted the production of volatile compounds by the oxidation of fatty acids, Maillard reaction, and Strecker degradation [61,62].

The numbers of volatile compounds in the HAD, VFD, and NAD groups were 54, 69, and 71, respectively (Figure 2a). As shown in Figure 2, aldehydes and alcohols were the primary compounds of dried *S. rugoso-annulata*. Aldehydes accounted for the highest percentage in the three groups, with a maximum value of 30.95% in the HAD group (Figure 2b), which may be attributed to the oxidation and degradation of UFA resulting in the generation of aldehydes [16]. Conversely, the highest content of aldehydes was found in the VFD group (Figure 2c), which may be due to differences in the amount of UFA among the three groups. Alcohols were the second-most abundant volatile compounds and the highest proportion showed in the VFD group. However, the lowest proportion and content of alcohols were found in the HAD group, indicating that alcohols may be unstable and easy to diffuse or degrade at high temperature [63].

According to the aroma description, there were a total of 41 volatile flavor compounds presented in *S. rugoso-annulata*. To further explore the differences and similarities of these components among the three groups, the PCA analysis, Venn diagram, and heatmap analysis were carried out (Figure 3). The PCA analysis was performed to investigate the distribution of volatile flavor components across three groups (Figure 3a). The first two principal components (PC), accounting for a total variance contribution of 87.7% (PC1 occupied 60%; PC2 occupied 27.7%), were used to explain the majority of the total sample information. The HAD group was situated on the negative direction of PC1, which was mainly positively correlated with 2-methybuanal, benzaldehyde, n-hexanol, ethyl acetate, acetic acid, octanoic acid, and 1-octen-3-ol. The VFD group was located in the upper right quadrant, which was mainly positively correlated with alcohols. Meanwhile, the NAD group was located in the bottom right quadrant, which was mainly positively correlated with aldehydes. The Venn diagram (Figure 3b) suggested that there were 22 common volatile flavor compounds among the three groups. Among the compounds, 1-hexanol, 1-octen-3-ol, benzaldehyde, and ethyl acetate were only detected in the HAD group. The higher content of benzaldehyde in the HAD group may be correlated with the lower content of phenylalanine, since the high temperature may promote the decomposition of phenylalanine into benzaldehyde [64]. Moreover, 3-octanol and undecane were detected only in the VFD group, while 2-heptanol was only detected in the NAD group. The heatmap analysis revealed the differences in the contents of volatile flavor components among the three groups, and it demonstrated that the VFD group showed greater similarity to the NAD group (Figure 3c). Hexanal, 2-octen-1-ol, and methyl caproate were the primary volatile flavor components of dried *S. rugoso-annulata*, giving it a fresh, fruit, citrus, vegetable, fatty, and sweet flavor. The volatile flavor compound with the highest content in HAD and VFD groups was hexanal, which was derived from n-6 polyunsaturated fatty acids and associated with a fresh, green, and fatty aroma [65]. Moreover, the volatile flavor compound with the highest content in NAD group was 2-octen-1-ol.

C8 compounds were the main volatile flavor compounds found in many mushrooms, which were derived from the oxidation of linoleic acid and linolenic acid [66]. There were eight C8 compounds detected in dried *S. rugoso-annulata*, including 1-octen-3-ol, 2-octen-1-ol, 3-octanol, 2-ethyl-1-hexanol, (E)-2-octenal, 3-octanone, 3-octen-2-one, and octanoic acid. Among these compounds, 2-ethy-1-hexanol was highest in VFD group, which was the only branched C8 compound; 1-octen-3-ol was found to be the main C8 compounds in other mushrooms [67], and it was only present in the HAD-treated *S. rugoso-annulata*. Similarly, a report indicated that the concentration of 1-octen-3-ol decreased with the increase in freeze-drying time [68]. However, *Lentinus edodes* dried by FD, HAD, and ND contained 103.78 μg/g, 5.72 μg/g, 105.30 μg/g, respectively [16]. Moreover, the content of octanoic acid in the HAD group was higher than the other groups, while 3-octanol was only detected in the VFD group, which correlated with the mushroom aroma. The contents of 2-octen-1-ol (green, citrus, vegetable, and fatty odor), (E)-2-octenal (mushroom, green, nut, and fat odor), and 3-octanone (mushroom, earthy, ketone, and resinous) were highest in the NAD group, while 3-octen-2-one, which is associated with a nutty aroma, had a higher concentration in both the VFD (99.00 ± 4.67 μg/g) and NAD (97.46 ± 29.72 μg/g) groups compared with the HAD group (14.19 ± 0.74 μg/g). 

In general, different drying methods had various effects on the volatile flavor compounds in *S. rugoso-annulata*. Among the three treatments, NAD treatment may be the optimal method for generating a high concentration of volatile flavor compounds in the mushroom.

### 3.7. Sensory Qualities Evaluation

The sensory quality of the mushroom was determined through the evaluation of five parameters: appearance, aroma, texture, intention to buy, and overall acceptability. The results were presented in a radar chart (Figure 4a) that clearly demonstrated consumers’ preference for the VFD group over the other two groups. With regard to appearance, a crucial factor that influences consumers’ purchasing decisions, there was found to be minimal discoloration in the VFD group when compared with the HAD and NAD groups (Figure 4b). However, the HAD group exhibited the most severe browning owing to the enzymatic browning and oxidation reactions promoted by high temperature [13].

## 4. Conclusions

In this study, the effects of different drying treatments (HAD, VFD, and NAD) on the nutrients and volatile compounds of *S. rugoso-annulata* were comprehensively analyzed. The results indicated that the drying treatments had significant impacts on the nutrients and volatile components of *S. rugoso-annulata*. VFD treatment was the optimal method for preserving nutrients, such as protein, fat, total amino acids, essential amino acids, MSG-like amino acids, sweet amino acids, monounsaturated fatty acids, and ergosterol. However, HAD treatment was beneficial for retaining total phenols, total flavonoids, lycopene, β-carotene, VD_2_, and antioxidant activity. Moreover, NAD treatment produced the most volatile and volatile flavor compounds. Further studies should focus on optimizing drying procedures to achieve the maximum preservation of both nutrients and volatile flavor components.

## Figures and Tables

**Figure 1 foods-12-02077-f001:**
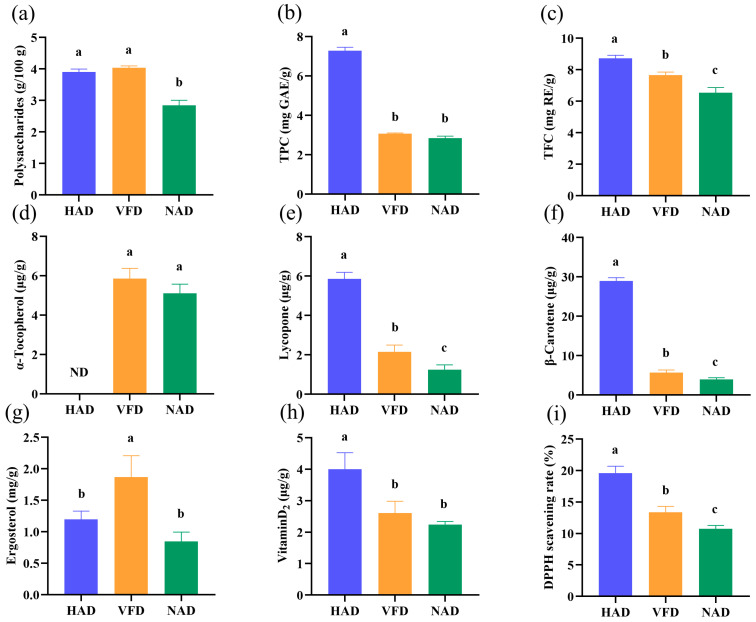
Contents of bioactive compositions and antioxidant activity in different-drying *S. rugoso-annulata*: (**a**) polysaccharides, (**b**) total phenolics, (**c**) total flavonoids, (**d**) α-tocopherol, (**e**) lycopene, (**f**) β-carotene, (**g**) ergosterol, (**h**) vitamin D_2_, and (**i**) DPPH scavenging rate. a–c: Different letters represent a significant difference (*p* < 0.05). ND: not detected. TPC: total phenolics content; TFC: total flavonoids; HAD: hot air drying; VFD: vacuum freeze drying; NAD: natural air drying.

**Figure 2 foods-12-02077-f002:**
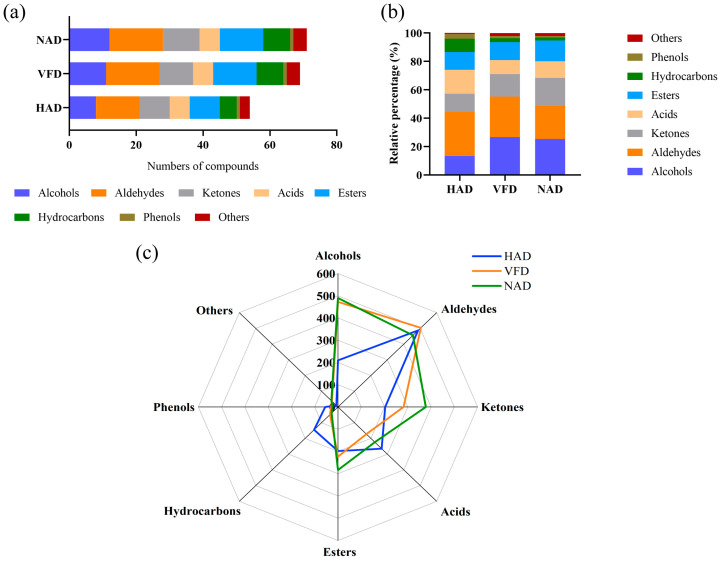
Types of volatile compounds in different-drying *S. rugoso-annulata*: (**a**) numbers of different types volatile compounds, (**b**) relative contents of different types volatile compounds, and (**c**) contents of different types volatile compounds. HAD: hot air drying; VFD: vacuum freeze drying; NAD: natural air drying.

**Figure 3 foods-12-02077-f003:**
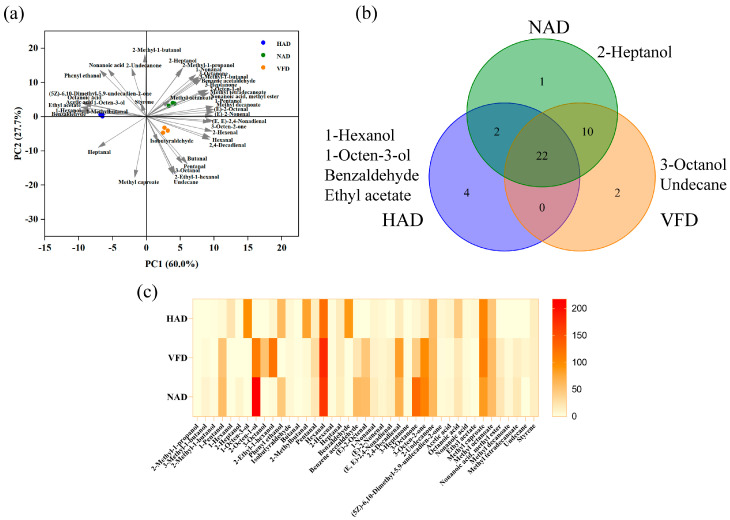
The differences and similarities of volatile flavor compounds in different-drying *S. rugoso-annulata*: (**a**) principal component analysis, (**b**) Venn diagram, and (**c**) heatmap. HAD: hot air drying; VFD: vacuum freeze drying; NAD: natural air drying.

**Figure 4 foods-12-02077-f004:**
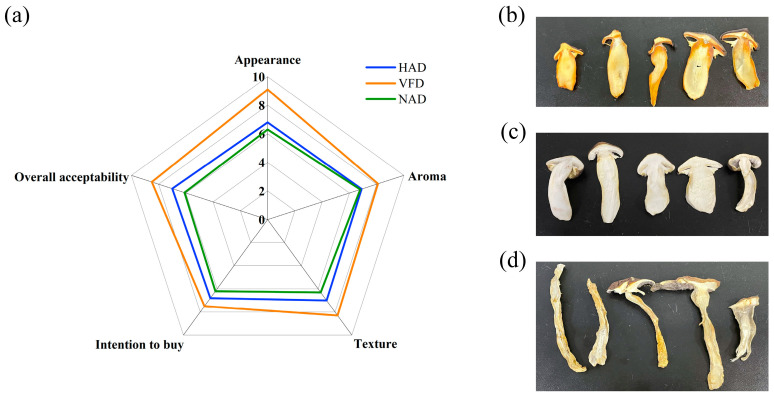
Sensory qualities of different-drying *S. rugoso-annulata*: (**a**) radar chart of sensory qualities evaluation, and (**b**–**d**) photos of HAD, VFD, and NAD *S. rugoso-annulata*. HAD: hot air drying; VFD: vacuum freeze drying; NAD: natural air drying.

**Table 1 foods-12-02077-t001:** Proximate compositions (percentage) in different-drying *S. rugoso-annulata*.

Compositions	HAD	VFD	NAD
Protein	17.99 ± 0.60 ^c^	25.67 ± 0.07 ^a^	22.69 ± 0.21 ^b^
Fat	0.80 ± 0.10 ^b^	1.12 ± 0.09 ^a^	1.05 ± 0.03 ^a^
Ash	7.16 ± 0.07 ^c^	9.46 ± 0.02 ^b^	11.26 ± 0.10 ^a^
Moisture	9.26 ± 0.11 ^a^	6.73 ± 0.17 ^b^	9.13 ± 0.23 ^a^
Total carbohydrate	64.80 ± 0.54 ^a^	57.03 ± 0.19 ^b^	55.88 ± 0.02 ^c^

^a–c^ Different superscript letters in the same row represent the significant difference (*p* < 0.05). HAD: hot air drying; VFD: vacuum freeze drying; NAD: natural air drying.

**Table 2 foods-12-02077-t002:** Amino acid compositions in different-drying *S. rugoso-annulata*.

Amino Acids		Content (g/100 g)		
		HAD	VFD	NAD
MSG-like	Asp	1.93 ± 0.06 ^c^	2.86 ± 0.14 ^a^	2.50 ± 0.22 ^b^
	Glu	3.11 ± 0.33 ^c^	5.05 ± 0.08 ^a^	4.59 ± 0.09 ^b^
	Total	5.04 ± 0.29 ^c^	7.92 ± 0.22 ^a^	7.09 ± 0.16 ^b^
Sweet taste	Thr	0.97 ± 0.10 ^c^	1.58 ± 0.05 ^a^	1.38 ± 0.01 ^b^
	Ser	1.10 ± 0.18 ^b^	1.68 ± 0.13 ^a^	1.63 ± 0.20 ^a^
	Ala	1.50 ± 0.11 ^b^	2.20 ± 0.06 ^a^	1.60 ± 0.01 ^b^
	Gly	0.96 ± 0.07 ^b^	1.37 ± 0.06 ^a^	1.25 ± 0.05 ^a^
	Pro	1.05 ± 0.05 ^a^	1.25 ± 0.08 ^a^	1.21 ± 0.22 ^a^
	Total	5.57 ± 0.36 ^c^	8.08 ± 0.37 ^a^	7.08 ± 0.27 ^b^
Bitter taste	Arg	0.29 ± 0.04 ^b^	1.28 ± 0.07 ^a^	1.42 ± 0.19 ^a^
	His	3.04 ± 0.30 ^b^	3.06 ± 0.19 ^b^	3.48 ± 0.08 ^a^
	Ile	0.97 ± 0.08 ^b^	1.20 ± 0.04 ^a^	1.02 ± 0.02 ^b^
	Leu	1.52 ± 0.12 ^c^	2.05 ± 0.05 ^a^	1.73 ± 0.03 ^b^
	Val	1.13 ± 0.10 ^c^	1.46 ± 0.06 ^a^	1.27 ± 0.05 ^b^
	Met	0.26 ± 0.02 ^b^	0.36 ± 0.02 ^a^	0.30 ± 0.03 ^b^
	Phe	0.88 ± 0.11 ^b^	1.27 ± 0.04 ^a^	1.13 ± 0.06 ^a^
	Total	8.09 ± 0.68 ^b^	10.67 ± 0.33 ^a^	10.35 ± 0.45 ^a^
Tasteless	Tyr	0.76 ± 0.08 ^c^	1.09 ± 0.05 ^a^	0.91 ± 0.06 ^b^
	Lys	0.84 ± 0.10 ^c^	1.83 ± 0.05 ^a^	1.54 ± 0.03 ^b^
	Total	1.60 ± 0.17 ^c^	2.92 ± 0.09 ^a^	2.45 ± 0.09 ^b^
other	Cys	0.01 ± 0.01 ^b^	0.08 ± 0.05 ^a^	0.06 ± 0.02 ^ab^
EAA		6.57 ± 0.59 ^c^	9.72 ± 0.25 ^a^	8.38 ± 0.22 ^b^
TAA		20.30 ± 1.46 ^c^	29.65 ± 0.91 ^a^	27.03 ± 0.50 ^b^

^a–c^ Different superscript letters in the same row represent the significant difference (*p* < 0.05). MSG-like: monosodium glutamate-like; TAA: total amino acids; EAA: essential amino acids, threonine, valine, methionine, isoleucine, leucine, phenylalanine and lysine; HAD: hot air drying; VFD: vacuum freeze drying; NAD: natural air drying.

**Table 3 foods-12-02077-t003:** Fatty acid compositions in different-drying *S. rugoso-annulata* (percentage).

Fatty Acids	HAD	VFD	NAD
C8: 0	0.49 ± 0.27 ^a^	0.95 ± 0.33 ^a^	0.76 ± 0.14 ^a^
C10: 0	8.03 ± 0.60 ^a^	2.00 ± 0.25 ^b^	2.14 ± 0.06 ^b^
C11: 0	0.51 ± 0.14 ^b^	0.43 ± 0.01 ^bc^	0.70 ± 0.14 ^a^
C14: 1	0.48 ± 0.13 ^a^	0.25 ± 0.04 ^a^	0.45 ± 0.05 ^a^
C16: 0	0.33 ± 0.03 ^b^	0.25 ± 0.03 ^c^	0.46 ± 0.02 ^a^
C16: 1	0.94 ± 0.05 ^b^	0.95 ± 0.03 ^b^	1.27 ± 0.02 ^a^
C18: 0	8.97 ± 0.41 ^a^	5.12 ± 0.11 ^c^	6.87 ± 0.13 ^b^
C18: 1	48.59 ± 1.58 ^b^	63.12 ± 0.92 ^a^	47.46 ± 0.45 ^b^
C18: 2	11.51 ± 0.54 ^a^	8.69 ± 0.20 ^b^	12.10 ± 0.35 ^a^
C18: 3	1.14 ± 0.03 ^b^	0.50 ± 0.02 ^c^	1.19 ± 0.01 ^a^
C20: 0	0.47 ± 0.03 ^a^	0.33 ± 0.01 ^b^	0.47 ± 0.01 ^a^
C20: 1	3.21 ± 0.15 ^a^	1.82 ± 0.08 ^c^	2.62 ± 0.42 ^b^
C20: 2	13.56 ± 0.90 ^b^	13.68 ± 0.82 ^b^	20.98 ± 0.09 ^a^
C20: 3	1.26 ± 0.03 ^c^	1.59 ± 0.04 ^b^	2.04 ± 0.01 ^a^
C22: 0	0.51 ± 0.16 ^a^	0.31 ± 0.01 ^b^	0.51 ± 0.05 ^a^
SFA (% of total FA)	19.30 ± 1.07 ^a^	9.39 ± 0.68 ^c^	11.90 ± 0.38 ^b^
MUFA (% of total FA)	53.22 ± 1.45 ^b^	66.13 ± 0.90 ^a^	51.79 ± 0.72 ^b^
PUFA (% of total FA)	27.48 ± 1.40 ^b^	24.78 ± 1.02 ^c^	36.31 ± 0.45 ^a^
UFA/SFA	4.19 ± 0.30 ^c^	9.69 ± 0.80 ^a^	7.41 ± 0.27 ^b^

^a–c^ Different superscript letters in the same row represent the significant difference (*p* < 0.05). SFA: saturated fatty acids; MUFA: monounsaturated fatty acids; PUFA: polyunsaturated fatty acids; UFA: unsaturated fatty acids; HAD: hot air drying; VFD: vacuum freeze drying; NAD: natural air drying.

**Table 4 foods-12-02077-t004:** The contents of mineral elements in different-drying *S. rugoso-annulata* (mg/kg).

	HAD	VFD	NAD
K	25,084.07 ± 1769.57 ^b^	36,321.73 ± 2818.04 ^a^	34,510.85 ± 1242.98 ^a^
Na	260.20 ± 43.25 ^a^	340.88 ± 44.16 ^a^	340.72 ± 41.04 ^a^
Ca	359.84 ± 9.40 ^b^	459.37 ± 27.84 ^a^	495.92 ± 63.26 ^a^
Mg	752.35 ± 13.43 ^c^	883.80 ± 36.40 ^b^	1049.52 ± 39.39 ^a^
Fe	130.94 ± 4.01 ^c^	209.09 ± 14.44 ^b^	282.07 ± 57.96 ^a^
Cu	4.47 ± 1.25 ^c^	18.72 ± 0.48 ^a^	7.06 ± 0.19 ^b^
Zn	16.15 ± 0.98 ^b^	44.75 ± 2.39 ^a^	16.50 ± 0.46 ^b^
Se	0.18 ± 0.01 ^b^	0.47 ± 0.03 ^a^	0.20 ± 0.01 ^b^
Cr	1.53 ± 0.10 ^a^	0.64 ± 0.03 ^b^	0.75 ± 0.03 ^b^
Co	0.12 ± 0.02 ^b^	0.18 ± 0.00 ^a^	0.14 ± 0.03 ^ab^
Cd	0.17 ± 0.08 ^b^	0.37 ± 0.05 ^a^	0.11 ± 0.01 ^b^
Hg	ND	0.012 ± 0.002	ND
Pb	0.11 ± 0.01 ^c^	0.22 ± 0.02 ^a^	0.15 ± 0.00 ^b^
As	0.52 ± 0.02 ^c^	3.65 ± 0.16 ^a^	1.04 ± 0.14 ^b^

^a–c^ Different superscript letters in the same row represent the significant difference (*p* < 0.05). ND: not detected. HAD: hot air drying; VFD: vacuum freeze drying; NAD: natural air drying.

**Table 5 foods-12-02077-t005:** The percentage contribution of essential trace elements to RDA in different-drying *S. rugoso-annulata*.

	RDA (mg/day)	Percentage Contribution (%) to RDA		
		HAD	VFD	NAD
Fe	8	10.80 ± 0.33 ^c^	17.25 ± 1.19 ^b^	23.27 ± 4.78 ^a^
Cu	0.9	3.28 ± 0.92 ^c^	13.73 ± 0.36 ^a^	5.18 ± 0.14 ^b^
Zn	11	0.97 ± 0.06 ^b^	2.69 ± 0.14 ^a^	0.99 ± 0.03 ^b^
Se	0.055	2.18 ± 0.13 ^b^	5.67 ± 0.37 ^a^	2.35 ± 0.07 ^b^
Cr	0.035	28.89 ± 1.92 ^a^	12.05 ± 0.51 ^b^	14.17 ± 0.54 ^b^
Co	0.0001	792.93 ± 102.15 ^b^	1164.61 ± 26.06 ^a^	953.75 ± 189.25 ^ab^

^a–c^ Different superscript letters in the same row represent the significant difference (*p* < 0.05). RDA: recommended daily allowances; HAD: hot air drying; VFD: vacuum freeze drying; NAD: natural air drying.

**Table 6 foods-12-02077-t006:** The estimated daily intake (EDI) and target hazard quotient (THQ) of potentially toxic trace elements in different-drying *S. rugoso-annulata*.

	EDI (mg/kg)				THQ				TI
	Cd (×10^−4^)	Pb (×10^−4^)	Hg (×10^−4^)	As (×10^−4^)	Cd	Pb	Hg	As	
HAD	0.187	0.121	ND	0.572	0.019	0.003	ND	0.191	0.212
VFD	0.407	0.242	0.013	4.015	0.041	0.006	0.004	1.338	1.389
NAD	0.121	0.165	ND	1.144	0.012	0.004	ND	0.381	0.398

ND: not detected. EDI: estimated dietary intake; THQ: toxic hazard quotient; TI: toxic index; HAD: hot air drying; VFD: vacuum freeze drying; NAD: natural air drying.

**Table 7 foods-12-02077-t007:** The contents of volatile compounds in different-drying *S. rugoso-annulata*.

Compounds		Aroma Description	Content (μg/g)		
			HAD	VFD	NAD
Alcohols	Trimethylsilanol	-	ND	23.57 ± 5.13 ^a^	14.90 ± 0.60 ^b^
	2-Methyl-1-propanol	Cocoa, almond	0.37 ± 0.22 ^b^	0.36 ± 0.07 ^b^	0.97 ± 0.32 ^a^
	3-Methyl-1-butanol	Alcoholic, cheese, pungent	ND	2.94 ± 0.75 ^b^	9.93 ± 1.58 ^a^
	2-Methyl-1-butanol	Wine, onion	1.93 ± 0.24 ^b^	ND	3.21 ± 0.59 ^a^
	1-Pentanol	Fruit, balsamic	5.13 ± 0.97 ^c^	45.44 ± 7.03 ^b^	58.67 ± 2.71 ^a^
	4-Ethylcyclohexanol	-	ND	1.07 ± 0.11 ^a^	0.91 ± 0.01 ^b^
	2,3-Dimethyl-1-butanol	-	ND	1.15 ± 0.15 ^b^	8.55 ± 2.89 ^a^
	1-Hexanol	Green, fatty, floral	24.43 ± 1.47	ND	ND
	2-Heptanol	Mushroom, arctic bramble, melon	ND	ND	7.50 ± 3.10
	1-Octen-3-ol	Mushroom, fatty, fruity, grass, perfumy, sweet	99.16 ± 1.80	ND	ND
	2-Octen-1-ol	Green, citrus, vegetable, fatty	ND	107.39 ± 5.03 ^b^	216.04 ± 6.08 ^a^
	3-Octanol	Mushroom	ND	53.86 ± 3.77	ND
	2-Ethyl-1-hexanol	Citrus, sweet, rose, green	7.13 ± 0.27 ^b^	117.60 ± 5.60 ^a^	3.02 ± 0.45 ^c^
	(E)-6-Methyl-2-hepten-4-ol	-	ND	105.17 ± 6.41 ^a^	103.78 ± 18.12 ^a^
	2,3-Octanediol	-	ND	ND	15.21 ± 0.70
	1-Nonen-4-ol	-	2.54 ± 0.05 ^b^	13.92 ± 2.72 ^a^	ND
	Phenyl ethanol	Floral, sweet, honey	69.11 ± 7.50 ^a^	ND	47.20 ± 0.84 ^b^
	Total alcohols		209.80 ± 5.69 ^b^	472.46 ± 13.43 ^a^	489.88 ± 21.60 ^a^
Aldehydes	Isobutyraldehyde	Pungent, malt, green	5.15 ± 0.44 ^a^	6.97 ± 3.60 ^a^	5.31 ± 0.82 ^a^
	Butanal	Pungent, green	0.59 ± 0.14 ^b^	1.68 ± 0.33 ^a^	0.86 ± 0.08 ^b^
	3-Methylbutanal	-	76.69 ± 7.03 ^a^	7.79 ± 1.69 ^b^	8.59 ± 2.07 ^b^
	2-Methylbutanal	Fruity, green, almond	85.54 ± 6.37 ^a^	4.04 ± 0.83 ^b^	4.29 ± 1.18 ^b^
	Pentanal	Almond, malt, pungent	18.22 ± 4.77 ^b^	36.83 ± 6.82 ^a^	21.84 ± 2.56 ^b^
	2-Methylpentanal	-	8.69 ± 0.05 ^b^	13.70 ± 2.51 ^a^	8.31 ± 1.12 ^b^
	Hexanal	Fresh, green, fatty	124.79 ± 4.61 ^c^	184.06 ± 5.62 ^a^	169.08 ± 1.48 ^b^
	2-Hexenal	Leaf, green, wine, fruit	ND	2.12 ± 0.32 ^a^	1.82 ± 0.08 ^b^
	Heptanal	Fat, citrus	16.91 ± 0.09 ^a^	16.37 ± 0.42 ^a^	15.32 ± 0.71 ^b^
	Benzaldehyde	Almond, burnt sugar	92.28 ± 4.18	ND	ND
	2-Ethyl-2-hexenal	-	8.95 ± 0.35 ^b^	20.31 ± 0.96 ^a^	1.73 ± 0.35 ^c^
	Benzene acetaldehyde	Green, sweat, phenolic	ND	19.49 ± 1.99 ^b^	59.37 ± 4.07 ^a^
	(E)-2-Octenal	Mushroom, green, nut, fat	ND	45.12 ± 4.33 ^b^	54.19 ± 3.68 ^a^
	1-Nonanal	Citrus, green, fat	8.08 ± 0.98 ^b^	9.70 ± 1.20 ^b^	14.73 ± 0.53 ^a^
	(E)-2-Nonenal	Cucumber, green, fat	5.23 ± 0.41 ^b^	7.52 ± 0.38 ^a^	7.74 ± 0.28 ^a^
	(E, E)-2,4-Nonadienal	Geranium, pungent	ND	12.02 ± 1.59 ^a^	12.16 ± 1.06 ^a^
	2,4-Decadienal	Seaweed	14.71 ± 1.83 ^c^	85.23 ± 3.24 ^a^	64.79 ± 2.50 ^b^
	2-Butyl-2-octenal	-	5.62 ± 0.47 ^b^	30.33 ± 3.56 ^a^	5.19 ± 0.33 ^b^
	Total aldehydes		471.45 ± 23.41 ^ab^	503.26 ± 14.78 ^a^	455.32 ± 8.01 ^b^
Ketones	2-Butanone	-	5.09 ± 1.54 ^a^	1.87 ± 0.30 ^b^	1.30 ± 0.23 ^b^
	3-Hexanone	-	ND	ND	0.77 ± 0.17
	3-Heptanone	Green, fruit	ND	0.99 ± 0.12 ^b^	2.05 ± 0.52 ^a^
	2-Heptanone	-	58.85 ± 3.74 ^a^	20.41 ± 1.88 ^b^	22.39 ± 4.49 ^b^
	3-Octanone	Mushroom, earthy, ketone, resinous	6.02 ± 0.19 ^c^	40.60 ± 1.80 ^b^	131.95 ± 6.55 ^a^
	3-Octen-2-one	Nut	14.19 ± 0.74 ^b^	99.00 ± 4.67 ^a^	97.46 ± 29.72 ^a^
	4,6-Dimethyl-5-hepten-2-one	-	35.03 ± 0.77 ^a^	35.26 ± 3.33 ^a^	30.17 ± 4.44 ^a^
	(E)-3-Nonen-2-one	-	6.67 ± 0.43 ^a^	7.24 ± 0.96 ^a^	5.32 ± 1.56 ^a^
	2-Undecanone	Floral, fruit	60.06 ± 2.15 ^a^	53.62 ± 6.32 ^a^	61.09 ± 0.99 ^a^
	2-(5-Oxohexyl)-cyclopentanone	-	9.90 ± 1.40 ^b^	20.14 ± 7.12 ^a^	21.76 ± 2.90 ^a^
	(5Z)-6,10-Dimethyl-5,9-undecadien-2-one	-	7.07 ± 0.96 ^a^	2.79 ± 0.51 ^b^	3.00 ± 0.14 ^b^
	Total ketones		202.87 ± 6.73 ^c^	281.92 ± 23.58 ^b^	377.27 ± 39.28 ^a^
Acids	Acetic acid	Pungent, vinegar	11.76 ± 1.50 ^a^	3.69 ± 0.41 ^b^	4.56 ± 2.42 ^b^
	Nonanoic acid	Green, fat	4.81 ± 0.25 ^a^	3.20 ± 0.01 ^b^	4.46 ± 0.56 ^a^
	Hexanoic acid	-	149.23 ± 12.82 ^b^	140.57 ± 1.89 ^b^	181.29 ± 23.00 ^a^
	Heptanoic acid	-	21.90 ± 3.42 ^a^	6.71 ± 2.50 ^b^	10.25 ± 2.53 ^b^
	Octanoic acid	Sweat, cheese	51.70 ± 6.96 ^a^	16.90 ± 4.46 ^b^	21.01 ± 2.91 ^b^
	2-Methylbutanoic acid	-	25.61 ± 3.39 ^a^	1.20 ± 0.96 ^b^	2.45 ± 0.77 ^b^
	Total acids		265.02 ± 15.62 ^a^	172.27 ± 5.39 ^c^	224.02 ± 26.70 ^b^
Esters	Methyl acetate	-	16.57 ± 4.63 ^a^	2.33 ± 0.62 ^b^	1.65 ± 0.44 ^b^
	Ethyl acetate	Pineapple	8.43 ± 2.94	ND	ND
	Methyl butyrate	-	1.32 ± 0.25 ^a^	0.70 ± 0.26 ^b^	0.49 ± 0.07 ^b^
	Methyl caproate	Fruit, fresh, sweet	102.87 ± 3.48 ^b^	114.63 ± 4.35 ^a^	86.42 ± 2.99 ^c^
	Hexyl formate	-	ND	1.37 ± 0.26 ^b^	1.86 ± 0.04 ^a^
	Methyl enanthate	-	16.07 ± 1.23 ^a^	7.99 ± 1.08 ^c^	9.88 ± 0.47 ^b^
	Methyl octanoate	Orange	52.30 ± 3.93 ^a^	54.08 ± 7.05 ^a^	56.29 ± 2.50 ^a^
	Nonanoic acid, methyl ester	Coconut	ND	12.22 ± 2.34 ^b^	22.96 ± 2.25 ^a^
	Methyl decanoate	Wine	ND	4.42 ± 0.38 ^b^	5.61 ± 0.61 ^a^
	2-Pentenoic acid, ethyl ester	-	ND	11.92 ± 4.92 ^a^	13.37 ± 2.11 ^a^
	Diethyl phthalate	-	0.79 ± 0.13 ^c^	3.89 ± 0.64 ^b^	66.46 ± 2.15 ^a^
	Methyl tetradecanoate	Orris	ND	9.63 ± 3.96 ^b^	18.33 ± 2.06 ^a^
	Total esters		198.34 ± 7.75 ^c^	223.18 ± 16.84 ^b^	283.33 ± 3.75 ^a^
Hydrocarbons	n-Pentane	-	4.28 ± 0.65 ^b^	6.20 ± 3.08 ^a^	2.11 ± 0.35 ^c^
	Methyl cyclohexane	-	0.41 ± 0.09 ^a^	0.53 ± 0.26 ^a^	0.84 ± 0.27 ^a^
	1-Ethyl-2-methylcyclopentene	-	ND	1.36 ± 0.10 ^a^	1.08 ± 0.03 ^b^
	Undecane	Green	ND	9.35 ± 1.24	ND
	9-Methyl-nonadecane	-	ND	1.95 ± 0.48 ^a^	1.20 ± 0.22 ^a^
	Heneicosane	-	0.52 ± 0.10 ^c^	2.13 ± 0.60 ^a^	0.92 ± 0.19 ^b^
	Toluene	-	126.17 ± 6.74 ^a^	8.20 ± 2.74 ^b^	10.77 ± 2.09 ^b^
	p-Xylene	-	ND	5.97 ± 1.21 ^a^	7.55 ± 0.79 ^a^
	Styrene	Balsamic, gasoline	13.77 ± 0.93 ^a^	12.94 ± 2.18 ^a^	13.96 ± 1.61 ^a^
	Total hydrocarbons		145.15 ± 7.14 ^a^	48.63 ± 3.40 ^b^	38.43 ± 5.15 ^c^
Phenols	3-Methyl-6-propyl-phenol	-	ND	29.22 ± 4.97 ^a^	23.58 ± 1.97 ^a^
	4-Ethyl-phenol	-	54.98 ± 5.79	ND	ND
	Total phenols		54.98 ± 5.79 ^a^	29.22 ± 4.97 ^b^	23.58 ± 1.97 ^b^
Others	Acetonitrile	-	3.65 ± 0.85 ^c^	24.28 ± 0.68 ^b^	28.49 ± 1.99 ^a^
	Dichloromethane	-	3.91 ± 1.93 ^a^	5.41 ± 2.15 ^a^	6.99 ± 2.84 ^a^
	Carbon disulphide	-	2.28 ± 0.66 ^a^	2.88 ± 2.04 ^a^	2.41 ± 0.77 ^a^
	1-Chloropentane	-	ND	4.25 ± 0.92 ^a^	2.03 ± 0.44 ^b^
	Total others		9.84 ± 3.33 ^b^	36.82 ± 4.26 ^a^	39.92 ± 6.03 ^a^
Total volatile flavor compounds			911.73 ± 33.13 ^c^	1196.80 ± 53.31 ^b^	1307.21 ± 47.60 ^a^
Total volatile compounds			1557.45 ± 55.29 ^c^	1767.76 ± 70.68 ^b^	1931.75 ± 90.18 ^a^

^a–c^ Different superscript letters in the same row represent the significant difference (*p* < 0.05). - means the compound has no aroma. ND: not detected. HAD: hot air drying; VFD: vacuum freeze drying; NAD: natural air drying.

## Data Availability

The data are contained within this article.

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
