# Peer review of "The Nutrients and Volatile Compounds in Stropharia rugoso-annulata by Three Drying Treatments"

_foods, 2023, doi:10.3390/foods12102077_

Round 1
Reviewer 1 Report
The manuscript entitled “The Nutrients and Volatile Compounds in Stropharia rugoso-annulata by three drying treatments” is well-written and has scientific merit. However, it needs to be revised according to the following comments:
Introduction:
The authors have mentioned some previous similar research on the drying of S. rugoso-annulata. However, the novelty of the present work needs to be clarified in contrast with the previous literature. Please mention the difference between the previous research and your current work.
Materials and Methods
-The Vacuum freeze drying (VFD) section needs to be rewritten with more precise information. The −50 °C might be the initial freezing step. Then during the primary and secondary drying steps, there needs to be a change in the temperature and pressure. Please mention these.
-Mushrooms are also a great source of B vitamins. Why didn’t you measure it?
Results and Discussion:
- What was the average drying time required to dry by these three processes? The total drying time required would be helpful in discussing the changes in the amounts of different compounds (especially antioxidants).
- You dried the Mushrooms and measured the proximate composition. So, it is necessary to mention the final moisture content in various dried samples first.
There are some grammatical errors in the manuscript. Please revise the entire manuscript thoroughly to check these.
Reviewer 2 Report
Dear Authors,
The work "The Nutrients and Volatile Compounds in Stropharia rugoso-annulata by three drying treatments" brings an interesting proposal and a great dataset. Below are listed some questions that will make the work even more relevant.
2.1. Mushroom samples: Add geographic coordinates.
2.2.3. Natural air drying (NAD): Add relative humidity of the environment.
2.8.1. Polysaccharides: How were these results expressed?
2.8.2 Total phenolics and total flavonoids: Extraction no agitation?
2.8.3. Ergosterol, vitamin D2 and -tocopherol: How were these results expressed?
2.8.4. -Carotene and lycopene: These results must take into account the sample, being expressed, for example, in microgram/100 g of sample.
Line 275-indicating that the level of MSG-like ingredients in S. rugoso-annulata was high (> 2 g/100g) [34] - Make clearer the importance of this comparison.
Lines 334-336: The recommended daily allowances (RDA) for Zn, 334 Cu, Fe, Se, Cr and Co are 11mg/day, 0.9 mg/day, 8 mg/day, 55 g/day, 35 g/day and 0.1 335 g/day, respectively [48,49]. Therefore, incorporating S. rugoso-annulata into one’s diet 336 can serve as a source of these crucial trace elements. - Describe the percentage contribution of these minerals to a commonly consumed portion of the mushroom. This can further reinforce the importance of product consumption.
Line 346-347: This may be due to contamination in the drying process. Why?
Why did the authors not identify the bioactive compounds? This would increase the relevance of the study.
